# Loss of the Novel Myelin Protein CMTM5 in Multiple Sclerosis Lesions and Its Involvement in Oligodendroglial Stress Responses

**DOI:** 10.3390/cells12162085

**Published:** 2023-08-17

**Authors:** Jiangshan Zhan, Yuanxu Gao, Leo Heinig, Malena Beecken, Yangbo Huo, Wansong Zhang, Pingzhang Wang, Tianzi Wei, Ruilin Tian, Wenling Han, Albert Cheung Hoi Yu, Markus Kipp, Hannes Kaddatz

**Affiliations:** 1School of Basic Medical Sciences, Peking University Health Science Center, Beijing 100191, China; zhanjs@bjmu.edu.cn (J.Z.); huoyangbo@bjmu.edu.cn (Y.H.); wangpzh@bjmu.edu.cn (P.W.); hanwl@bjmu.edu.cn (W.H.); achy@bjmu.edu.cn (A.C.H.Y.); 2Institute of Anatomy, Rostock University Medical Center, Gertrudenstraße 9, 18057 Rostock, Germany; leo.heinig@uni-rostock.de (L.H.); malena.beecken@uni-rostock.de (M.B.); markus.kipp@med.uni-rostock.de (M.K.); 3Center for Biomedicine and Innovations, Faculty of Medicine, Macau University of Science and Technology, Taipa, Macau 999078, China; gaoyuanxu@hsc.pku.edu.cn; 4Department of Medical Neuroscience, School of Medicine, Southern University of Science and Technology, Shenzhen 518055, China; 12133079@mail.sustech.edu.cn (W.Z.); weitz@sustech.edu.cn (T.W.); tianrl@sustech.edu.cn (R.T.); 5Department of Neurology, University Medical Center Rostock, University of Rostock, 18057 Rostock, Germany

**Keywords:** CMTM5, multiple sclerosis, cuprizone, experimental autoimmune encephalomyelitis, demyelination, endoplasmic reticulum stress

## Abstract

This study comprehensively addresses the involvement of the protein CKLF-like Marvel transmembrane domain-containing family member 5 (CMTM5) in the context of demyelination and cytodegenerative autoimmune diseases, particularly multiple Sclerosis (MS). An observed reduction in CMTM5 expression in post-mortem MS lesions prompted further investigations in both in vitro and in vivo animal models. In the cuprizone animal model, we detected a decrease in CMTM5 expression in oligodendrocytes that is absent in other members of the CMTM protein family. Our findings also confirm these results in the experimental autoimmune encephalomyelitis (EAE) model with decreased CMTM5 expression in both cerebellum and spinal cord white matter. We also examined the effects of a *Cmtm5* knockdown in vitro in the oligodendroglial Oli-neu mouse cell line using the CRISPR interference technique. Interestingly, we found no effects on cell response to thapsigargin-induced endoplasmic reticulum (ER) stress as determined by *Atf4* activity, an indicator of cellular stress responses. Overall, these results substantiate previous findings suggesting that CMTM5, rather than contributing to myelin biogenesis, is involved in maintaining axonal integrity. Our study further demonstrates that the knockdown of *Cmtm5* in vitro does not modulate oligodendroglial responses to ER stress. These results warrant further investigation into the functional role of CMTM5 during axonal degeneration in the context of demyelinating conditions.

## 1. Introduction

Multiple sclerosis (MS) is one of the most common demyelinating disorders affecting the central nervous system (CNS). Demyelination and axonal degeneration are pathological hallmarks found in MS lesions, leading to a heterogeneous degree of neurological deficits in individual patients [1]. At the pathophysiological level, demyelination (i.e., loss of myelin) may be triggered by inflammatory attacks on myelin by recruited peripheral immune cells as well as resident microglia. At the lesion sites, demyelinated axons are exposed to cytotoxic immune cells, among other factors, which may induce axonal damage. Non-inflammatory aspects such as the release of reactive oxygen species and glutamate, disruption of mitochondrial function, and accumulation of calcium may also contribute to the cytodegeneration of oligodendrocytes and progressive axonal degeneration [2]. Although it is known that an intact myelin structure has a predominant role in maintaining axon–myelin unit homeostasis, detailed characteristics of myelin components and their respective roles in maintaining axon–myelin function are still incompletely understood.

In recent years, members of the chemokine-like factor-like MARVEL-transmembrane domain-containing family proteins (CMTM) were found to be expressed as novel myelin proteins in the central and peripheral nervous system [3,4]. Under physiological conditions, CMTM6 is expressed on the adaxonal surface of Schwann cells and regulates axon diameters in the peripheral nervous system [3]. CMTM5, another member of the CMTM protein family, is highly enriched in mature oligodendrocytes of the CNS, and its depletion leads to impaired axonal integrity without affecting myelin biogenesis per se [4]. The expression and function of CMTM5 under pathological conditions such as in MS is, as yet, unclear.

In this study, we investigated the expression of CMTM5 in postmortem MS tissues as well as in different MS-related animal models. Functional studies using CRISPR interference were performed to study the relevance of *Cmtm5* expression in relation to cell responsiveness to endoplasmic reticulum (ER) stress.

## 2. Materials and Methods

### 2.1. Animals

For this study, 10-week-old C57BL/6 female and male mice were purchased from Janvier Labs (Le Genest-Saint-Isle, France). All experimental procedures were approved by the review boards for the care of animal subjects of the district government (District government of Mecklenburg-Western Pomerania, reference number 7221.3-1-001/19; District government of Bavaria, reference number 55.2-154-2532-73-15, Germany). At maximum, five animals were housed per cage (435 cm^2^ area). Animals were kept under standard laboratory conditions (12 h light/dark cycle, controlled temperature 23 °C ± 2 °C and 50% ± 5% humidity) with access to food and water ad libitum. The mice were allowed to acclimatize to the environment for at least one week prior to the beginning of the experiment. Body weights were controlled weekly. The mice were randomly assigned to the experimental group or control group, respectively (see Table 1).

### 2.2. Cuprizone Intoxication

In short, mice in the cuprizone experimental group were fed a diet containing 0.25% cuprizone (bis(cyclohexanone) oxaldihydrazone; Sigma-Aldrich, St. Louis, MO, USA) mixed into standard rodent chow (V1530–0; Ssniff, Soest, Germany). Cuprizone intoxication was performed as previously established by our group [5]. The cuprizone was weighed using precision scales and then mechanically mixed into ground standard rodent chow using a blender kitchen machine (Kult X, WMF Group, Geislingen an der Steige, Germany). The chow was mixed at low speed and under manual agitation of the entire machine for 1 min and was provided daily in two separate plastic Petri dishes per cage. The following exclusion criteria, which were checked daily, were applied to the mice in the animal experiment: severe weight loss (>10% within 24 h), severe behavioral disturbances (decreased locomotion, stupor, convulsions), or infections. During this study, no animal met the exclusion criteria.

### 2.3. Experimental Autoimmune Encephalomyelitis Induction and Disease Scoring

For the induction of experimental autoimmune encephalomyelitis (EAE), 10–12-week-old C57BL/6 female mice were subcutaneously immunized at two sites (0.1 mL/site, upper and lower back) with an emulsion of 1 mg/mL myelin oligodendrocyte glycoprotein (MOG_35–55_, peptide sequence MEVGWYRSPFSRVVHLYRNGK) dissolved in complete Freund’s adjuvant. After MOG_35–55_ immunization, each mouse was intraperitoneally injected with 300 ng pertussis toxin in 0.1 mL PBS twice, first on the day of immunization and then again the following day (Hooke Laboratories, Inc., Lawrence, KS, USA) as published previously [6]. To reduce stress after injection, mice were kept in their “home cage” without excessive noise or vibration. The severity of EAE development was scored as follows: 1: the entire tail falls over the observer’s finger when the mouse is picked up by the base of the tail; 2: the legs are not spread but are held close together when the mouse is picked up by the base of the tail, and the mice show a clearly visible wobbly gait; 3: the tail is flaccid, and the mice show complete paralysis of the hind legs (a score of 3.5 is assigned if the mouse is unable to raise itself when placed on its side); 4: the tail is flaccid, and the mice show complete paralysis of the hind legs and partial paralysis of the front legs, and the mouse barely moves in the cage but appears to be awake and eating; 5: the mouse is euthanized due to severe paralysis. Endpoints of EAE-mice were defined as a score of 4 for two consecutive days. Additional handlings/treatments were adopted according to the guideline for better implementation of the “three Rs” (replacement, reduction and refinement) to reduce suffering in EAE experiments [6]. When a mouse reached a score of 2.5, both control and EAE cages were supplied with food pellets and HydroGel (ClearH2O, Westbrook, ME, USA) on the floor of each cage for easier access. After an animal reached a score of 3, it received a subcutaneous injection of Ringer’s solution for fluidic supplement. In the presented EAE cohort, mice were sacrificed at 17 days post-immunization after the first mouse developed a score 4 and two mice developed a score of 3 (see Appendix A).

### 2.4. Tissue Preparation

Mice were deeply anaesthetized with ketamine (100 mg/kg i.p.) and xylazine (10 mg/kg i.p.) and then transcardially perfused with 20 mL of ice-cold PBS followed by a 3.7% formaldehyde solution (pH = 7.4). All immunohistochemical analyses were performed using paraffin-embedded 5 µm-thick coronal brain sections. For gene expression studies, the corpus callosum was manually dissected after perfusion with PBS only, immediately frozen and kept in liquid nitrogen until further processing.

### 2.5. Multiple Sclerosis Tissues

Paraffin-embedded post-mortem brain tissues were obtained via a rapid autopsy protocol from donors with progressive MS in collaboration with the Netherlands Brain Bank, Amsterdam. The study was approved by the Medical Ethical Committee of the Amsterdam UMC, the autopsy regimen and the ethical and legal declaration of the Netherlands Brain Bank were followed (coordinator Prof. I. Huitiga, https://www.brainbank.nl/about-us/ accessed on 14 July 2023), and all donors or their relatives gave written informed consent for the use of brain tissue and clinical information for research purposes. In total, 3 chronically active lesions from 3 donors and 4 non-MS control patients were included in this study (see Table 2). The mean age of the patients at death was 64.4 ± 18.3 years (mean ± standard deviation). The mean post-mortem delay was 7.8 ± 2.2 h. Lesions were classified via consecutive immunostaining for myelin proteolipid protein (PLP) and human leukocyte antigen [HLA]-DR (clone LN3), as previously reported [7].

### 2.6. Histological and Immunohistochemical Evaluation

For the histological evaluation, luxol fast blue (LFB)/periodic acid-Schiff (PAS) stains were performed to evaluate inflammatory demyelination within the white matter following standard protocols [8].

For the immunohistochemical studies, sections were deparaffinized, rehydrated, and if necessary, antigens were unmasked by cooking in Citrate (pH 6.0) buffer or tris(hydroxymethyl)aminomethane/ethylenediamine tetraacetic acid (Tris/EDTA) buffer (pH 9.0). After washing in PBS, unspecific binding sites were blocked by incubating the slides in the serum of the species in which the secondary antibody was raised for 1 h. Then, the sections were incubated overnight (at 4 °C) with the primary antibodies (see Table 3) diluted in 5% normal serum. Slides were then exposed to a PBS solution containing 0.35% hydrogen peroxide for 30 min in order to saturate endogenous peroxidase activities. After washing in PBS, the sections were incubated with biotinylated secondary antibodies for 1 h and then with a peroxidase-coupled avidin–biotin complex (ABC kit; Vector Laboratories, Peterborough, UK). The antigenic sites were then detected by incubation with 3,3′-diaminobenzidine (DAKO, Hamburg, Germany) for 10 min. Appropriate negative controls (omission of primary antibodies) were performed in parallel to ensure the specificity of the staining. Staining intensities were quantified via densitometrical analyses. All histological analyses were performed with coronal sections using a Leica DM6 B microscope equipped with the Leica DMC 6200 camera. Therefore, binary-converted images were evaluated within the ROI using ImageJ (NIH, Bethesda, MD [9]). A value of 100% represents maximum and 0% represents minimum staining intensity. The results are shown as staining intensity in (%) area of the entire ROI.

### 2.7. Generation of Cmtm5 Knockdown Oli-Neu CRISPRi Cell Line

An Oli-neu cell line [10] was lentivirally integrated with the plasmid cassette pMH0006 (pHR-SFFV-dCas9-BFP-KRAB) to generate the Oli-neu CRIPSRi cell line for further genetic engineering. pMH0006 was a gift from Martin Kampmann and Jonathan Weissman (Addgene plasmid # 135448; http://n2t.net/addgene:135448 accessed on 14 July 2023) [11]. The Oli-neu CRISPRi cell line was purified using fluorescence-activated cell sorting of Blue Fluorescent Protein (BFP). To generate a stable Cmtm5 knockdown cell line, sgRNA targeting Cmtm5 (sgRNA sequence: GAGCTGGGTGAAGCCCATCC) was cloned into the CRISPRia-v2 plasmid. CRISPRia-v2 was a gift from Jonathan Weissman (Addgene plasmid # 84832; http://n2t.net/addgene:84832 accessed on 14 July 2023) [12].

### 2.8. Real-Time RT-PCR

The primer sequences and individual annealing temperatures are given in Table 4. Relative quantifications of gene expression were performed for each sample using an internal standard curve generated by pooling cDNA from all samples. In this study, 18S expression levels were used as an internal reference. Gel electrophoresis and melting curves of the PCR products were routinely performed to determine the specificity of the PCR reactions. To exclude contamination of the reagents with either RNA or DNA, appropriate negative controls were performed (i.e., omission of RNA or cDNA; melting curve analyses, gel electrophoresis of the PCR products).

### 2.9. Single-Cell RNA Sequencing Analysis

Single-cell RNA (scRNA) sequencing data were obtained from public databases. Data from Wheeler et al. were accessed at GSE129609 [13]. Data from Jäkel et al. were accessed at GSE118257 [14]. We also downloaded the annotation data and UMAP coordination information from UCSC Cell Browser for Human scRNA sequencing analysis [15]. The analysis pipelines were detailed and described in a previous study [13]. Briefly, using Seurat [16], the expression matrices of samples were log-normalized, and doublets were removed. Canonical correlation analysis was performed to correct for batch effects and integrate different samples within each dataset. After integrating the data, principal component analysis was used to perform dimension reduction and clustering analysis. In all cases, the first 15 PCs were used. The cells were clustered using Louvain algorithm with a resolution parameter of 0.5. The MAST algorithm was used to conduct differential expression analysis for each cluster compared to all other cells. TSNE was used to visualize the data.

### 2.10. Statistical Analysis

All data are given as the arithmetic means ± Standard error of the mean (SEM). Differences between groups were statistically analyzed using the software GraphPad Prism 8 (GraphPad Software Inc., San Diego, CA, USA). The Shapiro–Wilk test was applied to test for normal data distribution. The definite statistical tests applied for the different analyses are provided in the respective figure legends. *p*-values of ≤ 0.05 were considered to be statistically significant. The following symbols are used to indicate the level of significance: * *p* ≤ 0.05, ** *p* ≤ 0.01, *** *p* ≤ 0.001, ns = not significant.

## 3. Results

### 3.1. Decreased Expression of CMTM5 Expression in the Cuprizone-Induced Animal Model of Demyelination

It has been suggested that CMTM5 is highly enriched in oligodendrocytes during maturation [4,17]. As a first step, we aimed to verify the expression of CMTM5 in an animal model of toxin-induced oligodendrocyte injury followed by demyelination. To this end, mice were continuously intoxicated with cuprizone for 1 week (1 wk), 3 weeks (3 wks) or 5 weeks (5 wks). Controls (Ctrl) were fed a normal diet throughout the entire experiment (i.e., animal cohort “Cup IHC/IF”, as described in the Materials and Methods Section). To verify the successful demyelination in the cuprizone model, coronal sections were processed for the analyses of anti-PLP labelling intensities. As shown in Figure 1a, profound demyelination of the medial corpus callosum was observed after 5 weeks of cuprizone intoxication,(Figure 1a: anti-PLP 81.28% ± 2.14% in Ctrl, 86.05% ± 2.01% at 1 wk Cup, 82.54% ± 2.30% at 3 wks Cup and 4.71% ± 0.67% at 5 wks Cup). Similar to the decrease in anti-PLP staining intensities during the course of cuprizone intoxication, the anti-CMTM5 staining intensities were found to be significantly reduced after 5 weeks in the cuprizone model (Figure 1a: anti-CMTM5 89.01% ± 0.76% in Ctrl, 84.65% ± 2.98% at 1 wk Cup, 83.33% ± 0.55% at 3 wks Cup and 34.15% ± 3.52% at 5 wks Cup). Of note, apart from widespread CMTM5-immunoreactivity morphologically resembling myelin sheaths, CMTM5-positive spheroids were observed in the medial corpus callosum of mice after 5 weeks of cuprizone intoxication (Appendix A).

To systematically evaluate the expression of CMTM proteins in the toxin-induced demyelination model, we analyzed the transcriptional expression of CMTM family genes using the RNA-seq dataset from control groups vs. 3 wks Cuprizone intoxicated groups (GSE213374 [18]). As shown in the heatmap (Figure 1b), only the expression of Cmtm5, but not other CMTM members, was reduced during the curprizone-induced demyelination process. To further confirm the transcriptome results, we analyzed the expression of Plp and Cmtm5 by real-time RT-PCR (Figure 1c: Plp, Control 100.00 ± 11.84% vs. 4 d Cup 6.817 ± 0.7011%; Cmtm5, Control 100.00 ± 7.068% vs. 4 d Cup 14.15 ± 1.630%; Figure 1d: Plp, Control 100.00 ± 6.909% vs. 3 wks Cup 11.88 ± 2.144%; Cmtm5, Control 99.99 ± 6.980% vs. 4 d Cup 65.30 ± 6.252%) using a set of independent biological samples (i.e., animal cohort “4 d Cup qPCR”, “3 wks Cup qPCR” as described in the Materials and Methods Section). Combined, these data demonstrate that CMTM5, as a novel myelin protein, is reduced after toxin-induced demyelination at both the mRNA and protein level.

### 3.2. CMTM5-Expression Is Reduced in the Inflammatory EAE Animal Model

The EAE model, in which encephalitogenic T helper 1 (Th1) and 17 (Th17) cells are induced to trigger autoimmune myelin attacks, is commonly used to study focal inflammatory demyelination, as also observed in acute MS lesions. To investigate the expression of CMTM5 in the inflammatory EAE model, acute MOG_35–55_ -EAE was induced in C57BL/6 mice (i.e., animal cohort “EAE” as described in the Materials and Methods Section). As shown in Figure 2a, mice were sacrificed at the peak of disease for the following analyses: To first validate inflammatory demyelination in the animal model, we performed real-time PCR on the acutely dissected cerebellum of EAE mice and age-matched controls. A significant induction of inflammatory chemokine Cxcl10 expression together with a reduction in Plp expression was observed in EAE animals (Figure 2b: Cxcl10, Control 100.00 ± 17.09% vs. EAE 2669 ± 877.5%; Figure 2c: Plp, Control 100.00 ± 9.756% vs. EAE 35.30 ± 6.652%). We then analyzed the expression of Cmtm5 in the EAE model. At the transcriptional level, a trend toward a decrease in Cmtm5 expression was detected in the cerebellum of EAE mice. (Figure 2d: Cmtm5, Control 100.00 ± 6.918% vs. EAE 63.55 ± 10.06%). Via bioinformatic analysis of the single-cell RNA sequencing (scRNA-seq) dataset from Wheeler et al. [13], we found that Cmtm5 was mainly enriched in cells of the oligodendrocyte lineage (see Figure 2h–m) and was reduced in the priming and acute phase of EAE (see Figure 2g). At the protein level, CMTM5 was found to be widely expressed in the white matter of the murine spinal cord (see Appendix A). A profound reduction in CMTM5 was observed in the white matter surrounding the inflammatory perivascular cuffs in the spinal cord of EAE mice (see Figure 2e,f). In addition, CMTM5-positive spheroids were present in the inflammatory foci of EAE spinal cord (see Appendix A). At the cellular level, CMTM5-positive spheroids in an advanced EAE-model were found near/within IBA1-positive microglia/macrophages (Appendix A) but not OLIG2-positive oligodendrocyte lineage cells (Appendix A).

### 3.3. A Cmtm5 Knockdown Does Not Affect Oli-Neu Cell Responsiveness to ER Stress

To further elucidate the function of CMTM5 in oligodendrocytes, we first confirmed endogenous expression of Cmtm5 in the mouse oligodendroglial Oli-neu cell line. The Oli-neu cell line was characterized via immunofluorescence staining with the oligodendrocyte marker anti-SOX10 (see Figure 3a). To perform loss-of-function manipulation of endogenic genes in the Oli-neu cell line, we lentivirally integrated the plasmid cassette (pHR-SFFV-dCas9-BFP-KRAB) into the Oli-neu cell line (i.e., Oli-neu CRISPRi). Thus, catalytically dead Cas9 (dCas9) fused with a KRAB transcriptional repression domain that blocks the transcription start sites in the genome, thereby inhibiting gene transcription [11]. sgRNAs targeting Cmtm5 were introduced into Oli-neu CRISPRi cell line using lentivirus, and the knockdown of Cmtm5 was confirmed using real-time PCR (Figure 3b: Cmtm5, non-targeting control 100.00 ± 2.497% vs. Cmtm5 sgRNA 10.63 ± 1.883%; Sox10, non-targeting control 100.00 ± 4.892% vs. Cmtm5 sgRNA 107.6 ± 3.210%). As recently shown by our group, ER-triggered cell death is involved in the cytodegeneration process of oligodendrocytes, and it is critical to maintain a dynamic stress response to ensure oligodendroglia survival under stress [19,20]. Since CMTM5 expression was found to be reduced in demyelination models (see Figure 1 and Figure 2), we asked whether knocking down Cmtm5 per se has an effect on the responsiveness of oligodendroglia to ER stress. To this end, we treated the Cmtm5 knockdown Oli-neu cell line with thapsigargin, an ER stress inducer that inhibits the sarco-endoplasmic reticulum Ca^2+^ ATPase. Our results show that the knockdown of Cmtm5 has no effect on the induction of Atf4, an activator of cytoprotective responses under cellular stress conditions [21] (Figure 3c: Cmtm5, non-targeting control DMSO treated 96.68 ± 4.840% vs. Cmtm5 sgRNA DMSO treated 13.78 ± 2.602%, non-targeting control thapsigargin treated 104.6 ± 2.564% vs. Cmtm5 sgRNA thapsigargin treated 11.17 ± 0.9641%; Atf4, non-targeting control DMSO treated 98.79 ± 6.265% vs. non-targeting control thapsigargin treated 414.1 ± 48.36%, Cmtm5 sgRNA DMSO treated 80.20 ± 4.478% vs. Cmtm5 sgRNA thapsigargin treated 547.1 ± 31.15%). In addition, no obvious morphological changes were observed in the Cmtm5 knockdown cell line when treated with thapsigargin. Similar to our findings, Buscham et al. reported that conditional knockout of Cmtm5 in mature oligodendrocytes does not affect oligodendroglial function, such as myelin biogenesis but functionally appears to contribute to progressive axonopathy, among other effects [4]. Overall, these results suggest that further investigations are warranted to understand the function of CMTM5 in the context of neuron–glia interaction.

### 3.4. CMTM5 Protein Expression Is Reduced in Demyelinated Post Mortem MS Lesions

As we have previously shown, CMTM5 expression was reduced in animal models recapitulating specific histopathological features of MS such as intrinsic oligodendrocyte degeneration (i.e., Cuprizone model) and inflammatory demyelination (i.e., EAE model) [22]. Therefore, in a translational approach, we next investigated whether the reduction in CMTM5 expression observed in animal models also occurs in lesions of progressive MS patients. To this end, brain sections from three progressive MS patients, together with four non-MS control patients, were processed for anti-PLP and anti-CMTM5 immunohistochemistry (see Table 2). Similar to the expression of PLP in human brains, CMTM5 was found to be enriched in the white matter and sparsely distributed in the cortex (see Figure 4b and Appendix A). In addition to the CMTM5-positive myelin sheath morphology in white matter and cortex, CMTM5-positive spheroids were observed in human brain white matter (see Appendix A, arrowheads). CMTM5, similar to PLP, was markedly reduced in the center of the lesion compared with normal-looking white matter near the lesion (Figure 4d,e: anti-PLP, 65.31% ± 3.79% in normal-appearing white matter (NAWM), 25.43% ± 0.19% in cortex, 23.92% ± 8.09% in the MS lesion center; anti-CMTM5, 41.73% ± 2.19% NAWM, 9.84% ± 4.38% in cortex, 14.44% ± 7.98% in the MS lesion center). To further identify the expressed cell type of CMTM5 in humans, we performed bioinformatic analysis using the published single nuclei RNA-sequencing dataset for MS [14]. Consistent with the cellular expression pattern in mouse brains, CMTM5 was predominately expressed in oligodendrocyte-lineage cells (see Figure 4j).

## 4. Discussion

In the present project, we were able to demonstrate that (i) CMTM5 expression is reduced in toxin-induced and inflammatory demyelination animal models; (ii) based on our in vitro experiments, knockdown of *Cmtm5* does not seem to affect cell responsiveness to ER stress, which is an early event during oligodendrocyte cytodegeneration; and (iii) that CMTM5 protein expression is reduced in post-mortem MS lesions.

As a first step, we investigated the expression of CMTM5 under the pathological condition of demyelination, taking into account its enrichment in mature oligodendrocytes at both the mRNA and protein level, as previously reported [4,23,24]. The high correlation of CMTM5 expression with PLP strongly suggests its property as a myelin protein in the central nervous system (see Figure 1 and Figure 4). Interestingly, the reduction in CMTM5 expression both at the protein level (about 40% left after 5 weeks of cuprizone intoxication) and at the mRNA level (about 60% left after 3 weeks of cuprizone intoxication) is less pronounced than for PLP, which is almost completely lost in the demyelinated regions (less than 10% left, see Figure 1c,d). This could be related to the CMTM5-positive spheroids detected in the demyelinated areas (see Appendix A). Interestingly, in the cuprizone model, we found a unique reduction in CMTM5 expression in oligodendrocytes that is absent in other CMTM members (see Figure 1b). There are several reasons why CMTM5 specifically should be studied in the context of demyelination: (i) *CMTM5* is independently localized on chromosome 14q11.2 compared to other CMTM members. Previous CMTM5-related studies have focused on the expression and function under pathological conditions such as tumorigenesis in particular [25,26]. However, little attention has been paid to the fact that CMTM5 is more abundant in the central nervous system. (ii) Other members of the CMTM family, such as CMTM6, which has been reported to be a myelin protein of the peripheral nervous system [3], were expressed at much lower levels and were not affected in CNS demyelination processes (see Figure 1b). Considering that CMTM5 is also expressed in Schwann cells and enriched in PNS myelin [3,27,28], studies on CMTM5 may elucidate novel functions of myelin proteins in both PNS and CNS. (iii) Compared to other CMTM members, CMTM5 does not contain an apparent chemokine-like structure. Therefore, it remains interesting to investigate how CMTM5 regulates neuron–glia interaction, particularly in the axon myelin unit, at the functional level. Further studies are necessary to investigate the dynamic regulation of CMTM5 expression, particularly its function in subsequent events such as activation and proliferation of oligodendrocyte progenitor cells and phagocytosis of myelin debris by microglia/macrophages in demyelinated lesions [22,29].

Structurally, CMTM5 belongs to the tetraspan–transmembrane proteins with small intracellular N- and C-terminal regions [30]. Other myelin tetraspan proteins, such as PLP [31], CD9 [32] and CD81 [33], have been shown to play pivotal roles in cell migration, vesicle trafficking and membrane adhesion [34]. Although lacking the N-terminal signal peptide, the secretory form of CMTM5 was found in the prostate lumen, secreted via a nonclassical secretory pathway [35]. Similarly, other CMTM members (e.g., CMTM-3, -4, -6, -7) have also been shown to regulate vesicle trafficking and stability of membrane proteins such as EGFR, VE-cadherin and PD-L1 [36,37,38,39,40]. Considering the enrichment of CMTM5 in the white matter, it remains intriguing to investigate whether CMTM5 is secreted into the cerebrospinal fluid and whether this results in properties as a biomarker or potential therapeutic approach.

Of note, in addition to the widespread expression of CMTM5 in the white matter, we found CMTM5-positive spheroids to be co-localized with markers for axonal damage (see Appendix A). In the context of our observation, it is interesting to note that Buscham et al. reported that *Cmtm5* deficiency induces progressive axonopathy [4]. Although oligodendrocytes are well known to provide trophic support to neurons in the axon–myelin unit, the specific mechanisms underlying axonopathy (e.g., interrupted axonal transport) due to myelin protein deficiency are still not entirely clear [2,4]. Frühbeis et al. showed that deficiency of different myelin proteins, such as PLP and CNP, impairs extracellular vesicle secretion [41]. Future studies need to investigate whether the absence of CMTM5 in the regulation of axonal integrity affects extracellular vesicle transport between oligodendrocytes and neurons. At the expressional level, the abundance of CMTM5 gradually increases during oligodendrocyte maturation [4,17]. Interestingly, we observed that *Cmtm5* is, compared to controls, reduced to around 10% after one week of cuprizone intoxication and recovered back to around 60% after 3 weeks of continuous cuprizone intoxication (see Figure 1c,d). Interestingly, the reduction in CMTM5 and PLP at the protein level became apparent only after 5 weeks of cuprizone intoxication (see Figure 1a). However, a reduction in mRNA expression of *Cmtm5* and *Plp* was already observed in mice intoxicated with cuprizone for only 4 days (see Figure 1c). The observed early reduction in *Cmtm5* and *Plp* expression in the cuprizone model may be due to regulated Ire1-dependent mRNA decay (RIDD), which is a selective decay of ER-bound mRNAs in response to ER stress [42]. According to the animal model characterization performed previously, cuprizone as a copper chelator first induces oxidative and ER stress in oligodendrocytes (e.g., within one week of cuprizone intoxication), which eventually leads to oligodendrocyte apoptosis [19,20]. Other degenerative processes, such as glial cell activation and axonal injury, follow with continuous intoxication (e.g., 3 weeks of cuprizone intoxication) [29]. These results imply that CMTM5, in addition to its role as a myelin component, might be involved in other reactive or degenerative cascades. Based on our and other studies, CMTM5 does not seem to be essential for myelin biogenesis [4] and OPC responsiveness to ER stress (see Figure 3c,d). Future experiments, for instance, using primary oligodendrocyte or co-culture with neurons, would be required to investigate the function of CMTM5 during oligodendrocyte maturation.

This study is not without limitations. First, our studies using the Oli-neu cell line examined the function of *Cmtm5* in oligodendroglial responses to ER stress induced by thapsigargin. However, the Oli-neu cell line was artificially produced as an immortalized mouse oligodendroglial cell line [10]. It is necessary to cross-compare the results with other oligodendroglial cell lines, such as OLN-93 and primary oligodendrocyte cultures. Second, the expression of ATF4 should be examined not only at the transcriptional level but also at the translational level to better assess the cellular responses under ER stress [21]. Third, although mice were randomly assigned to the experimental groups (i.e., Cup IHC/IF, 4 d Cup qPCR, 3 wks Cup qPCR, EAE), sex differences could be a potential confounding factor in the experimental setting. In addition, experimenters were not blinded to treatment/intoxication. However, the acquisition of IHC/IF results was always analyzed in a blinded manner by two independent evaluators and subsequently compared. In the cup IHC/IF cohort, mice were intoxicated with cuprizone for 1 week, 3 weeks, and 5 weeks, respectively. Control mice were sacrificed along with the mice that were intoxicated for 1 week. By adding multiple age-matched control groups, an even higher degree of comparability could be achieved.

## Figures and Tables

**Figure 1 cells-12-02085-f001:**
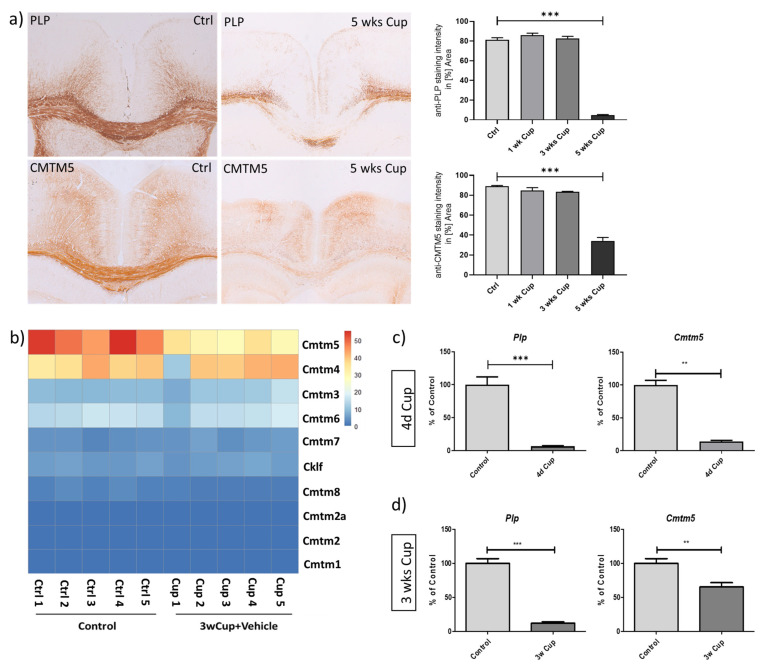
Reduction in CMTM5 expression in the cuprizone-induced demyelination model. (**a**) Representative images and quantitative evaluation of anti-PLP and anti-CMTM5 stained sections of the medial corpus callosum from control and cuprizone-intoxicated mice (*n* = 5). (**b**) RNA-seq heatmap analysis for transcript levels of CMTM family genes in control (*n* = 5) and 3 weeks cuprizone-intoxicated mice (*n* = 5) based on the dataset of GSE213374 [18]. Comparisons of the mRNA levels of *Plp* and *Cmtm5* between controls ((**c**), *n* = 7; (**d**), *n* = 8), 4 days ((**c**), *n* = 6) and 3 weeks ((**d**), *n* = 8) of cuprizone intoxication. ** *p* < 0.01, and *** *p* < 0.001 via Dunnett’s multiple comparisons test (**a**), and via Mann–Whitney tests (**c**,**d**). Scale bar: (**a**) = 200 µm.

**Figure 2 cells-12-02085-f002:**
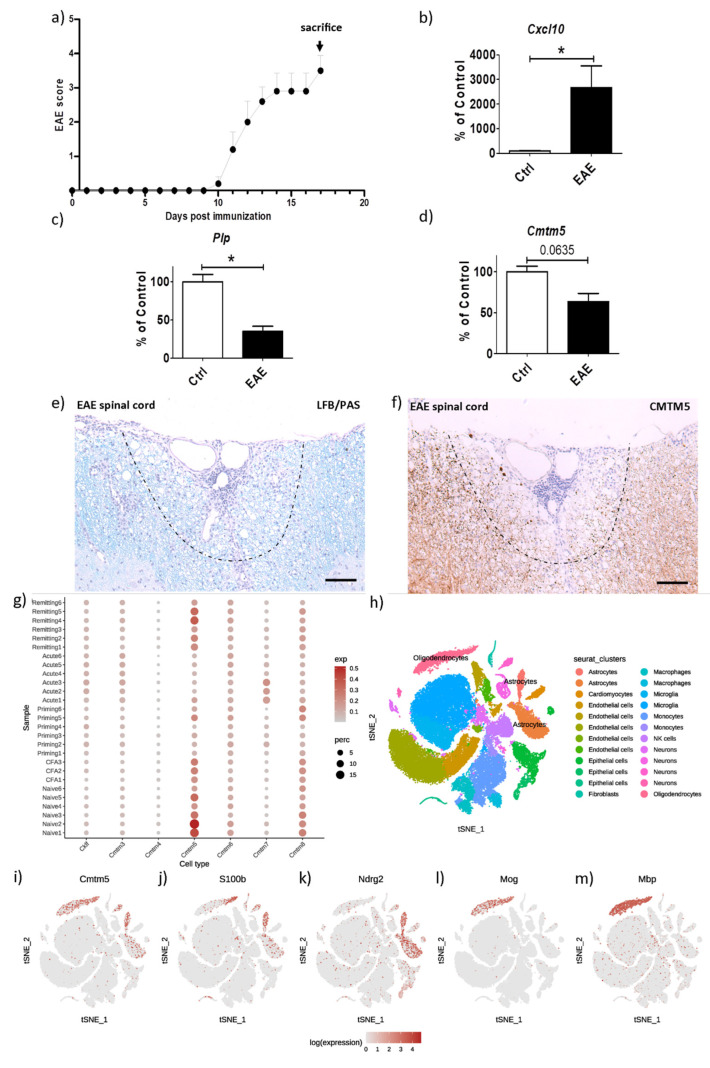
Reduction in CMTM5 expression in the inflammatory EAE model. (**a**) Clinical scores for MOG_35–55_-induced EAE in mice (*n* = 4). mRNA expression levels of (**b**) chemokine C–X–C motif ligand 10 (*Cxcl10*), (**c**) *Plp* and (**d**) *Cmtm5* in the cerebellum of control (*n* = 5) and EAE mice (*n* = 4). Representative images of (**e**) LFB/PAS stained and (**f**) anti-CMTM5 labelled sections of EAE spinal cord. The dashed line demarcates a representative demyelinated region in EAE spinal cord. (**g**) Dot plots showing a reduction in *Cmtm5* expression during the priming and acute phase of EAE mice after re-analysis of scRNA-seq dataset from Wheeler et al. [13]. Dot size corresponds to the percentage of nuclei expressing the gene in each cluster, the colour represents the average expression level (scale bars). (**h**) t-SNE clustering of the scRNA-seq dataset in 2D maps. (**i**–**m**) *Cmtm5* is predominantly expressed in oligodendrocyte-lineage cells based on differentially expressed genes for each cell type. * *p* < 0.05 via Mann–Whitney tests. scRNA-seq = single-cell RNA-sequencing. Scale bars: (**e**,**f**) = 30 µm.

**Figure 3 cells-12-02085-f003:**
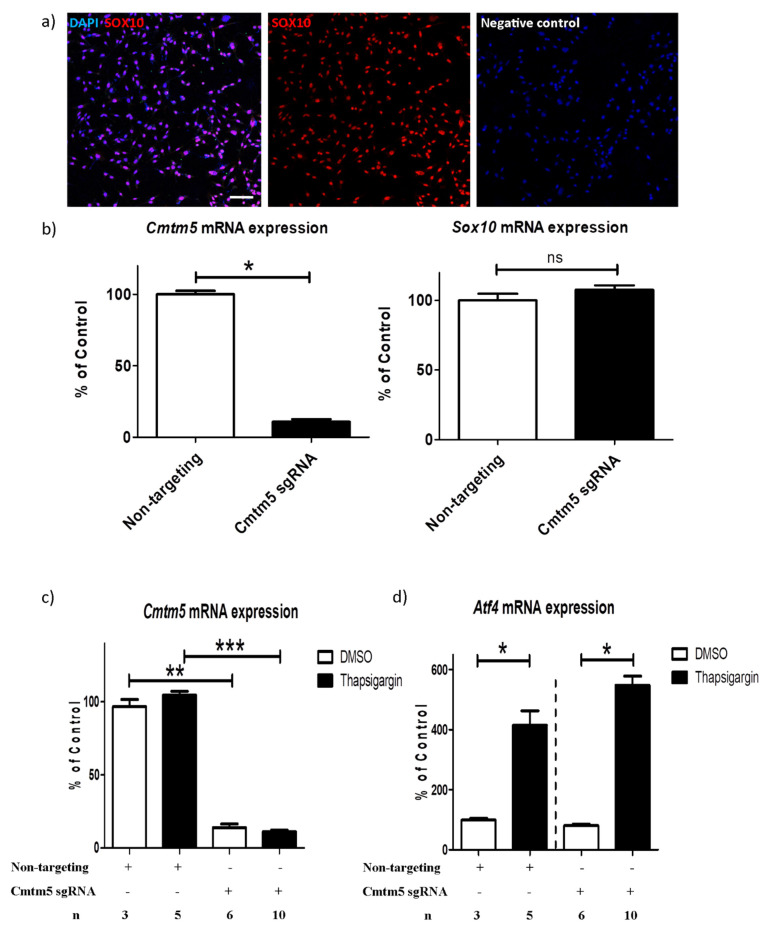
*Cmtm5* knockdown does not affect Oli−neu cell responsiveness to endoplasmic reticulum (ER) stress. (**a**) Characterization of the mouse OPC cell line Oli−neu by means of anti−SOX10 staining. (**b**) Stable and selective knockdown of *Cmtm5* in Oli−neu cell line carrying the CRISPR interference machinery without affecting the expression of *Sox10*. (**c**,**d**) Knockdown of *Cmtm*5 does not affect the induction of *Atf4* in Oli−neu when treated with the ER stress inducer thapsigargin (4 h; 1 µM). OPC = oligodendrocyte precursor cell. ER = endoplasmic reticulum. * *p* < 0.05, ** *p* < 0.01, *** *p* <0.001 and ns = not significant via Mann–Whitney tests. Scale bar: (**a**) = 100 µm.

**Figure 4 cells-12-02085-f004:**
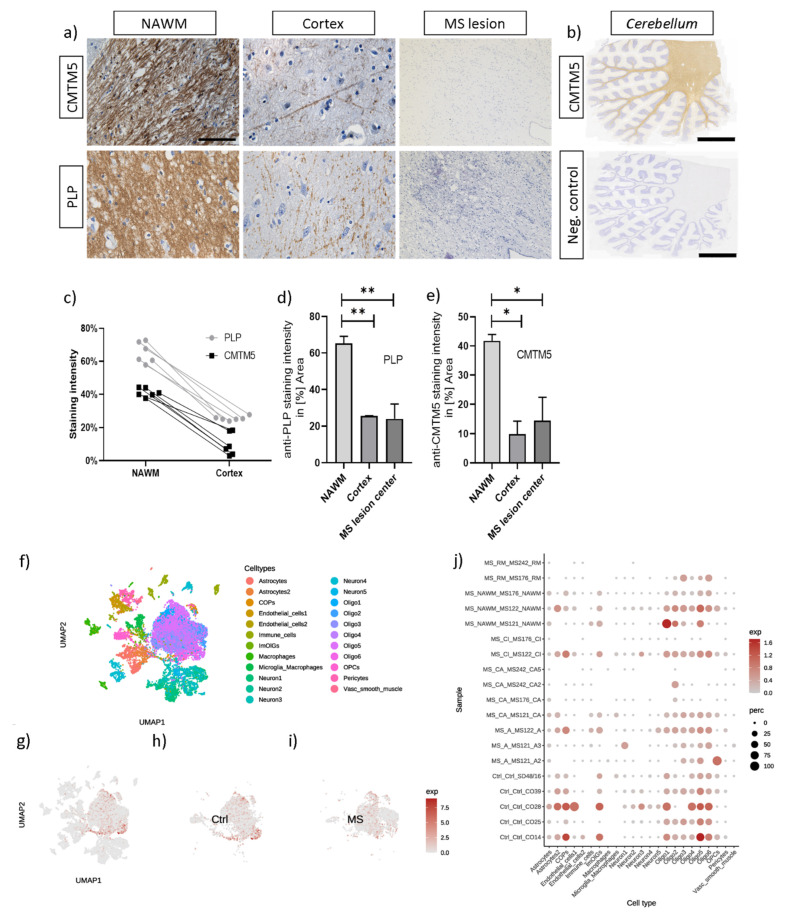
Reduction in CMTM5 expression in MS lesions. (**a**) Representative images and (**d**,**e**) densitometric analyses of anti-CMTM5 and anti-PLP labelled sections from chronic MS lesions (NAWM, *n* = 3 sections; Cortex, *n* = 3 sections; MS lesion center, *n* = 3 sections). Differences between groups were determined via Tukey’s multiple comparisons tests. Note the enriched expression of CMTM5 in the NAWM of MS patients (**a**) and fiber tracts of the cerebellum from non-MS human brains (**b**). (**c**) Visualization of PLP and CMTM5 optical density (staining intensity) in randomly selected areas in the NAWM (*n* = 6 images from 2 sections) or cortex (*n* = 6 images from 2 sections). (**f**–**i**) UMAP clustering of the snRNA-seq dataset in 2D maps after re-analysis of raw data from Jäkel et al. [14]. (**j**) *CMTM5* is predominantly expressed in oligodendrocyte-lineage cells based on the differentially expressed genes for each cell type. NAWM = normal-appearing white matter. UMAP = uniform manifold approximation and projection. snRNA-seq = single nuclei RNA-sequencing. * *p* < 0.05 and ** *p* < 0.01 via Tukey’s multiple comparisons tests. Scale bars: (**a**) = 100 µm, (**b**) = 5 mm.

**Table 1 cells-12-02085-t001:** Animal details of experimental groups used in this study. Abbreviations: wks, weeks; EAE, experimental autoimmune encephalomyelitis; ♂, male; ♀, female.

Cohort Name	Strain	Age	Sex	Experimental Group
Cup IHC/IF	C57BL/6	10 wks	♂	5 wks cuprizone (*n* = 5) 3 wks cuprizone (*n* = 5) 1 wk cuprizone (*n* = 5) Control (*n* = 5)
4 d Cup qPCR	C57BL/6	10 wks	♂	4 days cuprizone (*n* = 6) Control (*n* = 7)
3 wks Cup qPCR	C57BL/6	10 wks	♀	3 wks cuprizone (*n* = 8) Control (*n* = 8)
EAE	C57BL/6	11 wks	♀	EAE (*n* = 4), Control (*n* = 5)

**Table 2 cells-12-02085-t002:** Clinical details of MS patients. Abbreviations: ♂, male; ♀, female; h, hour; PMD, post-mortem delay; PPMS, primary progressive multiple sclerosis; SPMS, secondary progressive multiple sclerosis.

Sex	Age at Death in Years	MS Disease Duration in Years	Date of Birth	Date of MS Diagnosis	PMD	MS Type	Cause of Death
♀	35	10	1979	2004	10:20 h	SPMS	assisted suicide
♂	44	21	1965	1988	10:15 h	PPMS	End stage MS
♀	66	32	1945	1988	09:35 h	PPMS	assisted suicide
Controls							
♀	73	Not applicable	1929	Not applicable	04:00 h	Not applicable	Lung fibrosis
♀	60		1950		07:30 h		Infection
♀	86		1923		06:30 h		Multiple myeloma
♂	87		1918		06:32 h		Pneumonia

**Table 3 cells-12-02085-t003:** List of antibodies used in this study. Abbreviations: HIER, heat-induced epitope retrieval; RRID, research resource identifiers; Tris/EDTA, tris(hydroxymethyl)aminomethane/ethylenediamine. * Custom-made antibody provided by Prof. Hauke B. Werner.

Antigen	Used for	Species	Dilution	HIER	RRID	Supplier
PLP	Mouse and Human	Mouse	1:5000	None	AB_2237198	BioRAD, Hercules, CA, USA
CMTM5	Mouse	Rabbit	1:200	None	Custom made	*
CMTM5	Human	Rabbit	1:200	Citrate	AB_2928051	Abcam, Cambridge, UK
HLA-DR (LN3)	Human	Mouse	1:1500	Citrate	AB_10979984	Thermo Fisher, Waltham, MA, USA
SOX10	Mouse	Rabbit	1:80	None	AB_2927464	Abcam, Cambridge, UK
Synaptophysin	Mouse	Mouse	1:300	Citrate	AB_2198854	Abcam, Cambridge, UK
VGLUT1	Mouse	Mouse	1:1000	Citrate	AB_2923539	Abcam, Cambridge, UK
APP	Mouse	Mouse	1:5000	Tris/EDTA	AB_94882	Millipore, Burlington, MA, USA
OLIG2	Mouse	Mouse	1:200	Tris/EDTA	AB_10807410	Millipore, Burlington, MA, USA
IBA1	Mouse	Goat	1:200	Tris/EDTA	AB_870576	Abcam, Cambridge, UK

**Table 4 cells-12-02085-t004:** List of primers used in this study. Abbreviations: Cmtm5: chemokine-like factor-like MARVEL-transmembrane domain-containing protein 5. Plp: Proteolipid protein. Cxcl10: C-X-C motif chemokine ligand 10. Sox10: SRY-box transcription factor 10. Atf4: Activating transcription factor 4.

Gene Symbol	Sense	Antisense	T_a_/°C	Bp
*Cmtm5*	TGGTCTCCGTCTTTGCCTATG	CTCAGTGGTACTGGGCATCAG	62	68
*Plp*	TGGCGACTACAAGACCACCA	GACACACCCGCTCCAAAGAA	60	116
*Cxcl10*	CCAAGTGCTGCCGTCATTTTC	GGCTCGCAGGGATGATTTCAA	60	157
*Sox10*	AGGTTGCTGAACGAAAGTGAC	CCGAGGTTGGTACTTGTAGTCC	62	102
*Atf4*	GTTGGTCAGTGCCTCAGACA	CATTCGAAACAGAGCATCGA	60	109

## Data Availability

The data presented in this study are available upon request from the corresponding author.

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
