# Peer review of "Loss of the Novel Myelin Protein CMTM5 in Multiple Sclerosis Lesions and Its Involvement in Oligodendroglial Stress Responses"

_cells, 2023, doi:10.3390/cells12162085_

Round 1

Reviewer 1 Report

Legal Aspects

Please explain animal licenses here especially the listing of Oberbayern as no one oft he authors is affiliated there. Please also translate the German words to English.

Please also add 3R explanations to the method section considering the severity of your EAE experiment as well as the duration of days animals experienced a severe disease course.

Human studies: Please provide the ethic registration number.

Ethical aspects:

Using the word euthanasia for people with MS is in general not appropriate especially considering the presence of a German senior author.

Methods:

Mice were randomly assigned to a treatment that is great but treater and rater were not blinded. This should be standard. At least comment in limitations.

In this regards one control is not sufficient as treatment with Cuprizone was given for up to 5 weeks. At least comment in limitations.

Please comment on different animal sexes in the groups as major confounder.

Please indicate dosages of MOG, PTX used.

Figures:

Figure 1: can you explain difference in the time until decrease of CMTM5 and PLP in terms of staining compared to mRNA.

Figure 2: Please indicate n numbers e.g. EAE experiment. Please check this throughout the paper.

Discussion:

A section discussing the limitations of the work is missing. Please include this focusing on methods, statistics as well as translation to the human situation.

Author Response

We would like to take this opportunity to thank the reviewer for reviewing our work and contributing to the value of our study with his valuable comments and remarks. We hope we were able to answer all questions to the satisfaction of the reviewer.

Reviewer 2 Report

I have perused the manuscript by Zhan, Gao et al. with keen interest, and commend the authors on the comprehensive work undertaken. The robust methodologies adopted, encompassing RNA-seq, RT-PCR, immunohistochemistry, and lentivirus-mediated gene knockdown, significantly strengthen the study. The application of RNA-seq data offers an extensive view of the gene expression profile, further corroborated at an individual gene level through RT-PCR. Morphological aspects are explored via immunohistochemistry, while the gene knockdown study illuminates potential functional roles of CMTM5. The results indeed align seamlessly with the conclusion. I am inclined to recommend acceptance of the manuscript contingent on a few amendments.

  1. The title could be reshaped to more accurately encapsulate the principal findings. As an example, "The Impact of CMTM5 Depletion on Endoplasmic Reticulum Stress in Multiple Sclerosis Lesions and Animal Models" might be a suitable alternative. Nonetheless, the authors may devise a more fitting option.
  2. While the manuscript boasts exemplary data presentation and comprehensive discussion, the key findings have not been mirrored adequately in the abstract. I have composed a reworked version of the abstract that the authors might consider substituting for the current iteration, making necessary amendments as required: [This study comprehensively explores the involvement of CKLF-like Marvel transmembrane domain containing family member 5 (CMTM5) in cell responses to endoplasmic reticulum (ER) stress, within the context of neurodegenerative autoimmune diseases, specifically Multiple Sclerosis (MS). An observed reduction of CMTM5 expression in post-mortem MS lesions prompted further investigations in both in vitro and in vivo models. We identified a unique decrease in CMTM5 expression, absent in other CMTM members, in oligodendrocytes using the cuprizone MS model. Similarly, our findings corroborate these results in the Experimental Autoimmune Encephalomyelitis (EAE) model of MS, with diminished CMTM5 expression in both the cerebellum and the spinal cord's white matter. We also explored the impact of CMTM5 knockdown in the Oli-neu oligodendroglial mouse cell line via CRISPR interference. Interestingly, we found no effect on the cells' response to thapsigargin-induced ER stress, as determined by Atf4 activity, an indicator of cellular stress responses. Collectively, these results substantiate previous findings suggesting that CMTM5, rather than contributing to myelin biogenesis, is implicated in axonopathy. Our study further uncovers that CMTM5 does not modulate oligodendrocyte responses to ER stress. Future research should focus on other reactive or degenerative cascades to provide a more comprehensive understanding of CMTM5's role in myelin].
  3. The authors might consider further elucidation on what differentiates CMTM5 from other proteins within the CMTM family, and why specifically CMTM5 is worthy of investigation in the context of MS and demyelination.
  4. The authors should attend to the repetition of Reference 4 and 22.

In summary, this study constitutes a significant addition to the existing body of knowledge. Once the suggested modifications have been incorporated, the manuscript is poised to deliver considerable value to its readership.

Author Response

(The authors gave the same response as above.)

Reviewer 3 Report

The paper titled " Loss of the novel myelin protein CMTM5 in multiple sclerosis lesions and multiple sclerosis-related animal models is a novel finding. These findings motivate further research in CMTM5 in MS. Overall, the paper is well written and effectively communicates the main research findings. 

Author Response

(The authors gave the same response as above.)

Round 2

Reviewer 1 Report

Thank you all concerns are adressed. 

english is oke